# Multi-Label Fundus Image Classification Using Attention Mechanisms and Feature Fusion

**DOI:** 10.3390/mi13060947

**Published:** 2022-06-15

**Authors:** Zhenwei Li, Mengying Xu, Xiaoli Yang, Yanqi Han

**Affiliations:** College of Medical Technology and Engineering, Henan University of Science and Technology, Luoyang 471032, China; 200320221486@stu.haust.edu.cn (M.X.); yangxiaoli39@163.com (X.Y.); 210321221641@stu.haust.edu.cn (Y.H.)

**Keywords:** attention mechanisms, deep learning, feature fusion, image classification, fundus images

## Abstract

Fundus diseases can cause irreversible vision loss in both eyes if not diagnosed and treated immediately. Due to the complexity of fundus diseases, the probability of fundus images containing two or more diseases is extremely high, while existing deep learning-based fundus image classification algorithms have low diagnostic accuracy in multi-labeled fundus images. In this paper, a multi-label classification of fundus disease with binocular fundus images is presented, using a neural network algorithm model based on attention mechanisms and feature fusion. The algorithm highlights detailed features in binocular fundus images, and then feeds them into a ResNet50 network with attention mechanisms to extract fundus image lesion features. The model obtains global features of binocular images through feature fusion and uses Softmax to classify multi-label fundus images. The ODIR binocular fundus image dataset was used to evaluate the network classification performance and conduct ablation experiments. The model’s backend is the Tensorflow framework. Through experiments on the test images, this method achieved accuracy, precision, recall, and *F*1 values of 94.23%, 99.09%, 99.23%, and 99.16%, respectively.

## 1. Introduction

Fundus images are used by ophthalmologists and computer-aided diagnostics to detect fundus disease such as diabetic retinopathy, glaucoma, age-related macular degeneration, cataracts, hypertension, and myopia. Ophthalmologists have progressively adopted computer-aided diagnosis as its accuracy has increased in recent years. The system assists doctors in making partial diagnoses and saves both doctors and patients time and effort [1,2,3].

Early detection of fundus disease is critical for patients to avoid blindness. Abnormalities in the fundus can indicate different types of disease when a single fundus image is analyzed in three color channels. Patients usually develop ocular diseases differently in each eye due to the complexity and mutual independence of ocular diseases. Figure 1 shows right and left fundus images, taken from the ODIR dataset [4], of a patient with diabetic retinopathy and myopia in the right eye but not the left. The majority of fundus image research focuses on segmenting fundus structures or detecting anomalies in certain fundus diseases [5,6]. As a result, the ability to classify the whole range of disease on fundus images is critical for the development of future diagnosis systems.

Various algorithms in the fields of enhancement [7], segmentation [8,9,10], and classification [11,12] of fundus images have been developed based on merging image processing and deep learning principles. Deep learning algorithms can be sufficiently trained and are less prone to overfitting for datasets with more images, and test results accuracy can exceed 95%. The fundamental issue in classifying multi-label fundus images is the insufficient data, which prevents the model from being effectively trained. The second is that fundus images with more obvious lesions, such as glaucoma and other disorders that develop later before more obvious lesions appear, are easier to identify, and classification accuracy is significantly lower.

This research offers an algorithm based on a ResNet attention mechanisms method to fuse the features of binocular fundus images for classification problems. The system takes binocular fundus images as input and adds an attention mechanism to classify disease in a global, multi-label approach. By applying enhancement processing to the original image, the problem of low classification accuracy of fundus images with obscure abnormalities is solved. Using data augmentation and batch processing methods to train the data can solve the problem of uneven sample distribution.

### 1.1. Classification of Fundus Images

Most fundus image classification challenges nowadays are focused on identifying a single disease with or without conditions such as diabetic retinopathy [13], myopia [14], glaucoma [15], age-related macular degeneration [16], and other eye disorders. Gour et al. [17] used a single fundus image and developed a convolutional neural network using a transfer learning model to achieve high classification accuracy for multi-labeled images. SGD was used to optimize the network and improved the training set accuracy from 85.25% to 96.49%. However, the classification accuracy is low for fundus images containing glaucoma; one of the reasons is that the dissimilarity of early lesions in these diseases is not significant and not easily detected during classification. Second, the dataset has significantly less data than other disease images, making the model sensitive to overfitting when classifying these diseases. Joon Yul Choi et al. [18] discovered that the number of classes has a significant impact on classification performance. The VGG-19 network was used in this study to classify three types of fundus images, and the accuracy fell to 41.9% when the number of classes was increased to five. As a result, the critical problems that should be solved as soon as possible are how to handle the test dataset so that it is equally distributed and how to train a high-performance neural network to increase the classification accuracy of fundus images for each disease class.

### 1.2. Image Augmentation

A major challenge is averaging positive and negative sample distributions and enhancing image quality to increase classification accuracy. The number of input modules in a classification model impacts how well the network performs. The problem of unequal image distribution is common with multi-label data. The data upsampling method, in which the images are rotated, flipped, cropped, and other operations to augment the dataset with insufficient samples. The transfer learning method, in which the weight parameters are obtained by training on large ImageNet image datasets, and it is easier to obtain optimal results when using the pre-trained weights. Luquan et al. [19] improved the accuracy from 62.82% to 75.16% using transfer learning, but the model is prone to overfitting for image classes with small datasets. Third, by changing the underlying network, the model can perform better, even with small samples, Wang et al. [20] used Vgg16 to classify multi-labeleed fundus images with an accuracy of 86%, and changing to EfficientNetB3 improved the accuracy to 90%.

### 1.3. Attention Mechanisms

Image augmentation solves the problem of unequal sample distribution, but complicated lesions in the fundus, such as microaneurysms and hemorrhages, remain hard to identify. The shallow neural network learns the image’s texture features; as the network deepens, it learns the image’s semantic information. The rich semantic information can improve the network’s classification performance. Including the attention module allows the image to properly learn the spatial position information of lesions. This module imitates humans in finding significant regions in complicated situations and has applications in a variety of vision tasks [21], including image classification, target identification, image segmentation, and facial recognition. As indicated in the correlations in Figure 2, it may be split into six types based on the data domain: channel, spatial, temporal, and branching attention mechanisms, as well as channel and spatial attention and spatial and temporal attention mechanisms. Hu et al. [22] proposed the SENet channel attention network, which includes a squeeze-and-excitation (SE) module at its foundation. The SE module can gather data information, capture inter-channel relationships, and enhance the representation. However, it has the disadvantage of being unable to capture complex global information and having a high model complexity. Sanghyun Woo et al. [23] proposed the convolutional block attention module (CBAM) to improve global information exploitation. It connects the channel attention and spatial attention mechanisms, allowing the network to focus on features and their spatial locations. CBAM can also be added to any existing network architecture due to the network’s lightweight design.

### 1.4. Model Optimization

When a high-performance model is employed as the basic classification network, the total performance metric improves [24]. This is because the network’s prediction capacity is strongly related to its recognition of the features of the fundus image. To process the input fundus images, a high-performance neural network must be used. Multi-model fusion enables the creation of two models for extracting feature vectors, which will then be joined by vectors during fusion to increase classification accuracy. Wang et al. [20] analyzed model fusion-based classification of binocular fundus images using EfficientNet to extract features and then input the features into a classifier for classification, a two-stage classification technique with 90% accuracy. In general, the deeper the network, the better the classification. However, the deeper the network, the greater the risk of overfitting. ResNet extends network optimization by incorporating a residual module, which increases classification performance [25,26]. Furthermore, in deep learning, dataset size is important in determining classification performance. Pre-training weights loaded on huge ImageNet datasets are then trained on the target dataset to acquire the appropriate training parameters via transfer learning. Gour et al. [17] utilized neural networks trained through transfer learning to train binocular fundus images. The classification accuracy for cataract disease was 97% and that for glaucoma disease was 54%, with considerable variation in classification accuracy resulting in lower confidence in the model.

To address the above problems, this paper focuses on the design of a fundus image classification network with binocular feature fusion based on the attention mechanism. To solve the network overfitting problem caused by insufficient data, image enhancement and augmentation processes are performed on the original fundus images, and the training efficiency of the model is improved using transfer learning. The features of binocular fundus images were extracted by fusing ResNet and attention modules for the classification task, which increased the network’s ability to handle details. In subsequent experiments, the effectiveness of the model is verified by comparative analysis and ablation experiments.

## 2. Materials and Methods

In order to perform efficient classification of fundus diseases, a Binocular Fundus Photographs Classifying Network (BFPC-Net) based on attention mechanism and feature fusion is proposed, as shown in Figure 3. The BFPC-Net is composed of 3 parts: image augmentation model (IAM), residual attention module (RAM), and feature fusion module (FFM). The network has two main characteristics: (1) the addition of a residual network and attention mechanism fusion module, which causes the network to pay more attention to lesion feature information and improves the feature difference between lesions and background; (2) the addition of a multi-model fusion module, which combines binocular fundus images to determine the type of disease and improve classification accuracy.

The BFPC-Net takes a patient’s binocular fundus images as input and outputs six disease classes, one normal class, and one other class. To average the number of inputs in each class, the BFPC-Net begins with an image augmentation module. To extract shallow features, the number of image channels is enhanced by utilizing 3 × 3 convolutional layers, batch normalization, and ReLU activation layers. The RAM module is the residual attention module, which enhances network depth and overcomes the gradient disappearance and gradient explosion problems. Since binocular fundus images are used, the features extracted by the two networks need to be fused and then passed through the ReLU activation layer, with dropout and fully connected layers used to obtain the classification results.

### 2.1. Fundus Image Dataset

The ODIR dataset is from the Peking University International Competition on Ocular Disease Intelligent Recognition (ODIR-2019), which includes label information and retinal fundus images.

Figure 4 shows the distribution information of each label. The dataset consists of 6392 images; 44.95% were normal fundus images, 55.05% were diseased fundus images, while 25.16% of the diseased fundus images were from patients with diabetic retinopathy.

Table 1 shows the distribution of original images and preprocessed images for each type of label after balancing the data samples.

The experimental evaluation strategy is divided into two parts: first, the model’s overall accuracy (accuracy, A), precision (precision, P), recall (recall, R), and *F*1 values are assessed for all classes. Second, the same evaluation of the above metrics is performed for each class. The comparative study of the integrated metrics offers a foundation for evaluating the model’s overall performance.
(1)$$\mathrm{Accuracy}=\frac{TP+TN}{TP+FP+TN+FN}$$
(2)$$\mathrm{Precision}=\frac{TP}{TP+FP}$$
(3)$$\mathrm{Recall}=\frac{TP}{TP+FN}$$
(4)$$F1_{score}=\frac{2TP}{2TP+FP+FN}$$
where *TP* denotes a positive label and a positive predicted value, *FP* denotes a negative label and a positive predicted value, *TN* denotes a negative label and a negative predicted value, and *FN* denotes a positive label and a negative predicted value.

### 2.2. Image Augmentation Module

The larger the image, the more texture and detail it contains, and the better the features that can be captured. However, the classification performance peaks when the image size exceeds a particular threshold, and the computational cost increases as the image increases. This algorithm is intended for the image augmentation module (IAM), which includes three components: image normalization, image-weighted enhancement, and data augmentation. Details of the image preprocessing are shown in Figure 5.

The image-weighted enhancement can be formulated as Equation (5).
(5)$$I_{weight}=I_{org}*\alpha +I_{blur}*\beta +\gamma$$
where α=4, β=−4, γ=128, $I_{org}$ is the original image, and $I_{blur}$ represents the blurred image after convolution of the original image with the Gaussian kernel. The Gaussian blurring step is shown in Equation (6).
(6)$$I_{blur}=I_{org}*kernal_{h\times w}$$

The size of the Gaussian kernel is h=w=63, and the standard deviation of the values in the h direction and w direction is 10.

The primary steps in the data augmentation process are follow-on rotation, left/right inversion, and up/down flip. Figure 6 depicts the results. The use of data augmentation can help to alleviate the difficulties of overfitting and low classification accuracy caused by unbalanced data distribution.

### 2.3. Residual Attention Module

The model combines the residual architecture with the attention mechanism to extract deep semantic information from binocular fundus images, and the architecture is shown in Figure 7. The image augmentation feature map $F\in R^{\left(C\times H\times W\right)}$ is given as input. In order to focus on the feature mapping relationship between the channels of the image, the channel attention map is $U\in R^{\left(C\times 1\times 1\right)}$. The spatial attention map $U^{\prime }\in R^{\left(C\times H\times W\right)}$ is added to obtain the spatial relationship of the local region. A residual architecture is designed to improve network depth by merging the inputs F and $U^{\prime }$, which are then activated by ReLU to produce the output $M\in R^{\left(1\times H\times W\right)}$. Woo [20] et al. used a sigmoid activation function, but we adopted the ReLU function as the activation function in our research, which improved the model’s generalization ability.

In the channel attention module, three convolutional layers are set up to capture the nonlinear relationships of features between channels. $F\in R^{\left(C\times H\times W\right)}$ as input, sequentially infers a 1D channel attention map, yielding the channel features $F^{\prime }\in R^{\left(C\times H\times W\right)}$.

The channel attention is designed to capitalize on the inter-channel relationship of features. Average-pooling has been commonly used to aggregate spatial information. Sanghyun Woo [23] use the average-pooling and max-pooling gathers information about distinctive object features to infer finer channel-wise attention. We argue that average-pooling efficiently learns the extent of the target object, but max-pooling misses some information. Two distinct spatial context descriptors are produced by average-pooling processes. These two descriptors are utilized in the dense layer to reduce feature dimensionality, and the result is fed into the nonlinear activation layer to get the output at $U\in R^{\left(C\times H\times W\right)}$. As shown in Equation (7):(7)$$U=\sigma \left(FC\left(AvgPool\left(F^{\prime }\right)\right)+FC\left(AvgPool\left(F^{\prime }\right)\right)\right)\times F^{\prime }$$
where AvgPool(•) denotes global average pooling, FC(•) denotes full connection, and σ(•) denotes ReLU activation.

The spatial attention module focuses on the location information of the fundus lesion. Zagoruyko et al. [27] demonstrated that the superimposed pooling operation effectively highlighted the feature region. The input to the spatial attention module is *U* and the output is *U′*. In short, the spatial attention is computed from Equation (8):(8)$$U^{\prime }=Conv(MaxPool(AvgPool(U)))\times MaxPool(AvgPool(U))$$
where Conv(•) denotes convolution and MaxPool(•) denotes global maximum pooling.

The input of the module that merges the residual architecture with the attention mechanisms is the combination of the preprocessed image and the attention module output, as described in Equation (9):(9)$$M=U^{\prime }+F$$

### 2.4. Feature Fusion Module

Binocular fundus images typically contain one or more diseases, and detecting simply monocular fundus images cannot provide a comprehensive analysis of a patient’s condition. The deep learning training process is a way of finding the global optimal solution, and hyperparameters, such as learning rate, tend to cause the model to fall into a local optimal solution at a particular point, leading the model to stop optimizing. Multi-model fusion strategies can contribute to the reduction of this problem. The model output features are merged to get better outcomes by constructing a multi-model feature fusion module FFM based on binocular features.

As calculate in Equation (10), a sliding average method is used to combine multi-model characteristics:(10)$$p=\frac{1}{T}\sum {}_{\left(i=1\right)}^{T}w_{i}y_{i}$$
where T denotes the number of models and is set to 2, $w_{i}$ denotes the weights of individual models and is set to 0.5, 0.5 respectively, and $y_{i}$ denotes the predicted values.

### 2.5. Loss Function

To reduce data jitter and increase model performance on the test set, the predicted values are processed using the sliding average method in the label smoothing method when training the model. As shown in Equation (11):(11)$$y_{k}^{\prime }=(1-v)\times y_{k}+\frac{v}{K}$$
where $y_{k}^{\prime }$ denotes the *k*th label value after smoothing, $y_{k}$ denotes the true value of the *k*th label, *v* denotes the error rate and takes the value of 0.1, *K* denotes the number of classes, and the total number of class labels is 8.

The loss function after the label smoothing process can be calculated as:(12)$$Loss(y_{k}^{\prime },\hat{y_{k}})=-\sum {}_{\left(k=1\right)}^{K}y_{k}^{\prime }lg(\hat{y_{k}})$$
where $\hat{y_{k}}$ denotes the probability of the predicted classes.

## 3. Results and Discussion

### 3.1. System Specifications

The model was trained and tested using the Keras deep learning framework, on a Windows 10 Pro system with an NVIDIA CUDA 10.2 for GPU acceleration. The BFPC-Net was built using Tensorflow as the back-end Keras framework, using the Adam optimizer for loss reduction, with a learning rate of 0.001, and an epoch of 100. The configuration of the hyperparameters are shown in Table 2.

### 3.2. Experimental Results and Discussion

#### 3.2.1. Analysis of Experimental Results for the ODIR Dataset

Figure 8 and Figure 9 show the model fit while training the model to evaluate the classification performance of BFPC-Net. As shown in the figures, the model fits fast to the optimum and there is no overfitting of the training and validation sets on the model. Table 3 shows that when comparing the performance metrics of [20,28,29] research in classifying fundus images, MCGS-Net has lower accuracy and recall and does not classify disease images effectively. ResNet enhances the depth of the network for more detailed fundus images for analysis, and the classification performance is better than the above two models. EfficientNet is a lightweight network with fewer model parameters that improves the relevant metrics, but the metrics are still low. With fewer samples, the BFPC-Net proposed in this research can quickly fit the model and achieve high accuracy.

In the case of class imbalance, the high or low performance metric for each class better reflects the overall performance of the model. Figure 10 shows the BFPC-Net classification performance on eight of the classes in the ODIR dataset. The method achieves improved classification results in the dataset and can produce better classification results for classes with less images.

#### 3.2.2. Analysis of Ablation Experiments Results

Using images with image sizes of 256 × 256 and 128 × 128 for training, the classification results are shown in Figure 11. The results show that, when other conditions are the same, the image is larger and the classification results are better.

Even with 128 × 128 images as input, the trained accuracy, precision, and recall all reach 0.8942, 0.9665, and 0.9660, respectively, which shows that the classification effect of the model for low pixel images is equally excellent.

The BFPC-Net was separated into the basic network (Baseline) and Baseline + FFM models for independent experiments to validate the contribution of the IAM module and the FFM module to the model overall, and the results are shown in Table 4. The results show that adding the FFM module improves accuracy by 13.14% and the other three performance measures by 15.7%, 20.87%, and 18.37%, respectively.

#### 3.2.3. Model Performance Analysis

As shown in Table 5, when comparing the model to the research [30] method in terms of accuracy and parameters, the BFPC-Net improves the accuracy by 5.52% and the *F*1 value by 10.45%, with fewer parameters than the VGG16 model used in the research [30].

## 4. Conclusions

Using the ODIR dataset, a deep convolutional neural network architecture for binocular fundus image classification is proposed and evaluated. By simply inputting patients’ binocular fundus images, the method can yield fundus disease classifications with high confidence. The experimental results show that BFPC-Net overcomes the problems of a small fundus image dataset and low disease classification accuracy by combining image enhancement, residual attention, and feature fusion modules. BFPC-Net can provide a comprehensive treatment plan for patients by combining their binocular fundus images.

In the future, more types of fundus diseases images can be used to classify, especially rare diseases in clinic. The difficulty of this type of problem is that the training effect of small sample data classification is poor. However, the classification of such diseases is more suitable for clinical applications.

## Figures and Tables

**Figure 1 micromachines-13-00947-f001:**
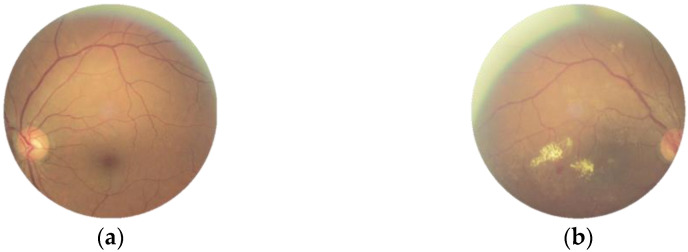
Images of the ODIR dataset. (**a**) No disease. (**b**) Diabetic retinopathy and myopia.

**Figure 2 micromachines-13-00947-f002:**
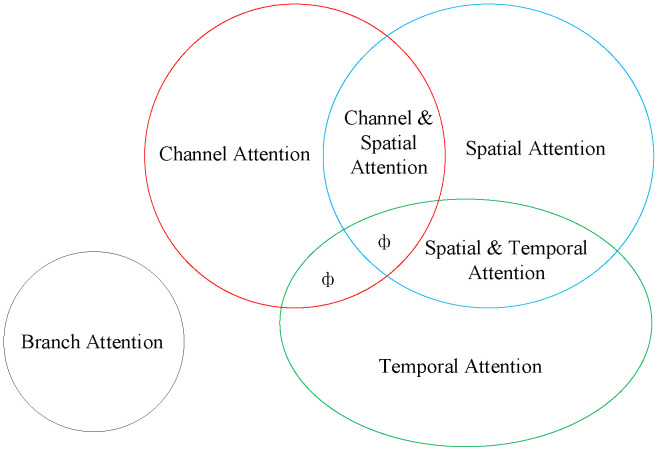
Classification of attentional mechanisms (ф indicates no relevant classification).

**Figure 3 micromachines-13-00947-f003:**
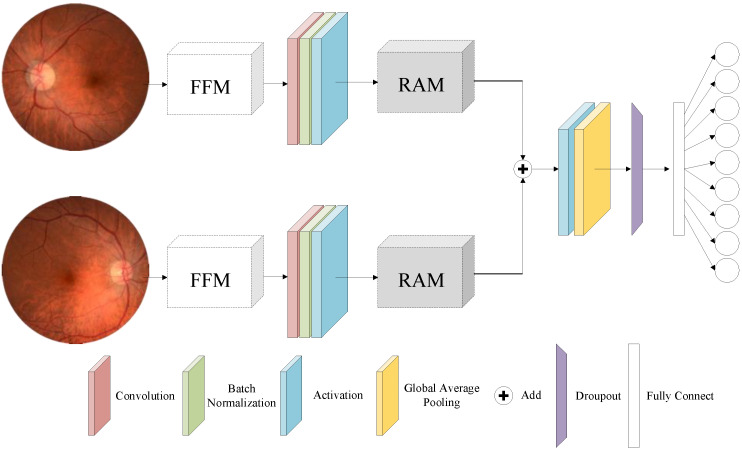
BFPC-Net model. IAM: image augmentation model; RAM: residual attention module; FFM: feature fusion module.

**Figure 4 micromachines-13-00947-f004:**
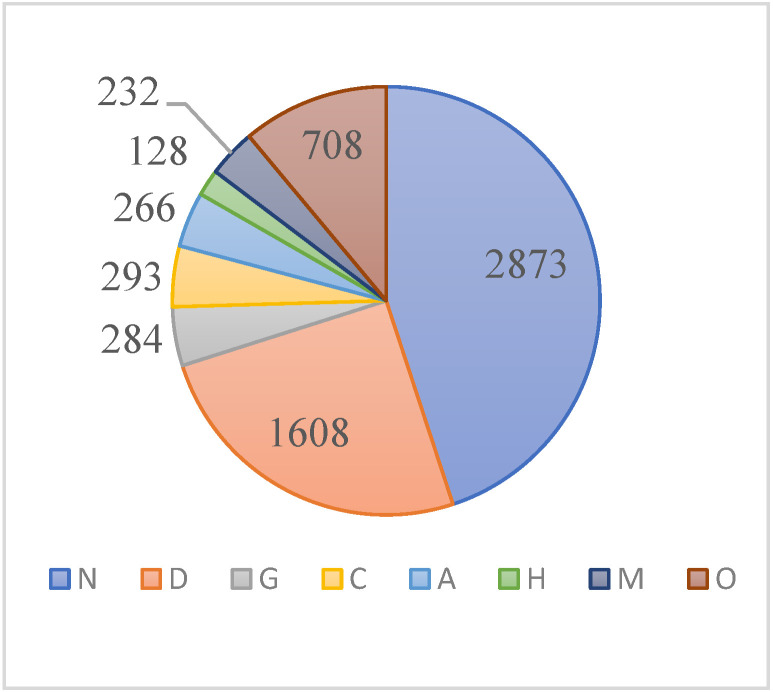
Distribution of ODIR dataset. Normal (N), diabetes (D), glaucoma (G), cataract (C), age-related macular degeneration (A), hypertension (H), pathological myopia (M), other diseases/abnormalities (O).

**Figure 5 micromachines-13-00947-f005:**
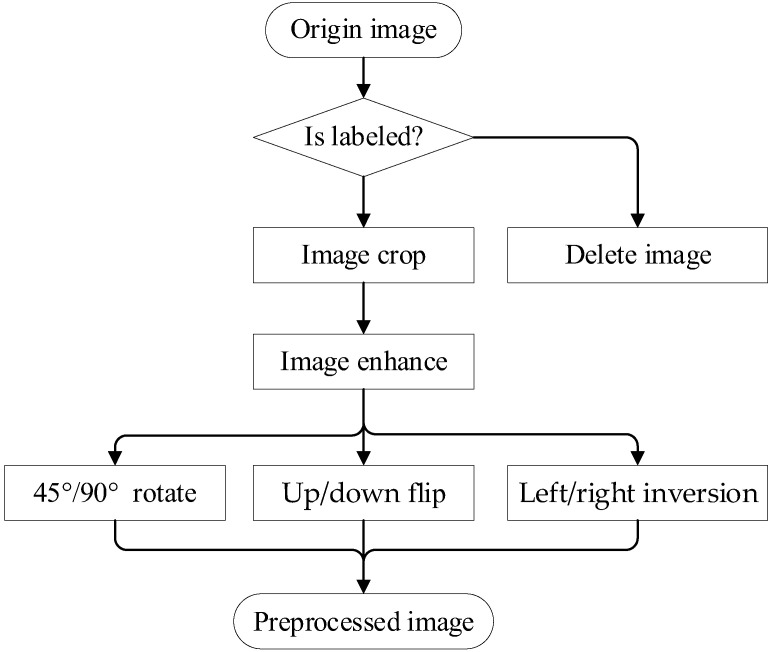
Image preprocessing flow.

**Figure 6 micromachines-13-00947-f006:**
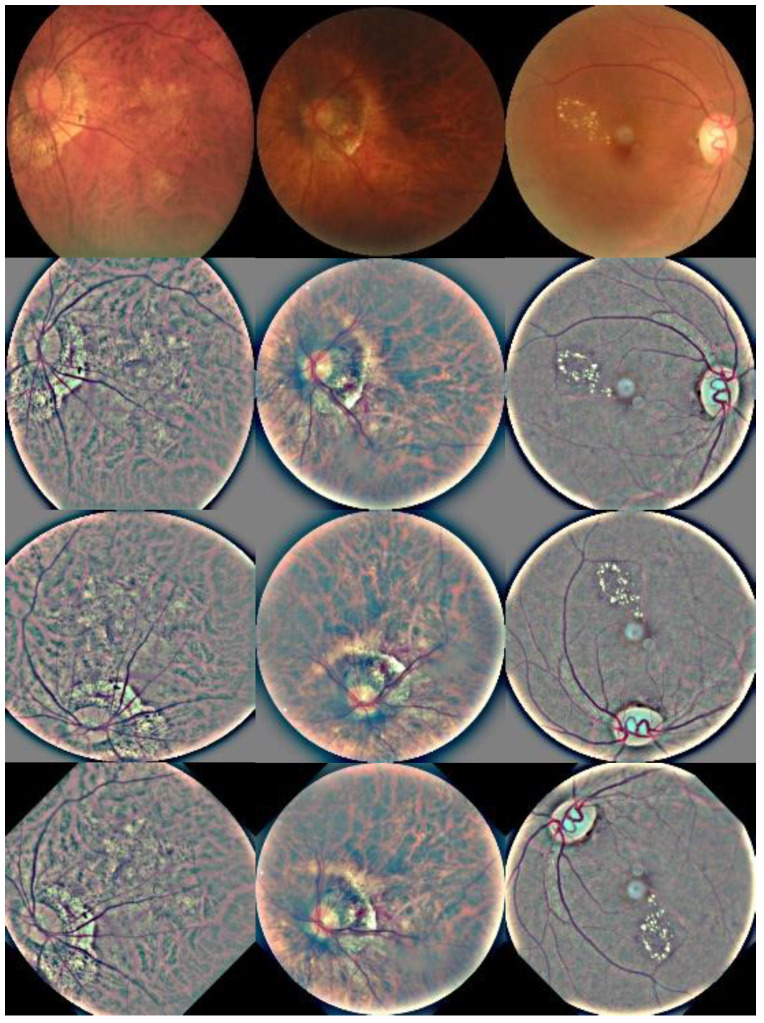
Image augmentation results.

**Figure 7 micromachines-13-00947-f007:**
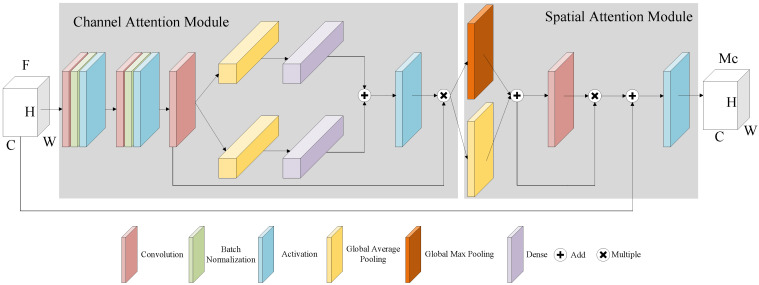
Residual attention module.

**Figure 8 micromachines-13-00947-f008:**
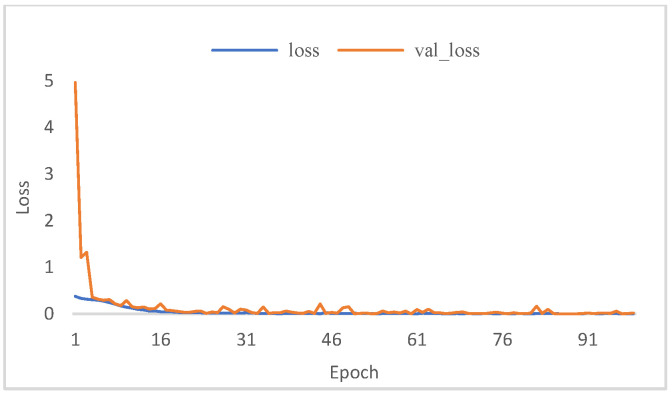
BFPC-Net training set and validation set losses.

**Figure 9 micromachines-13-00947-f009:**
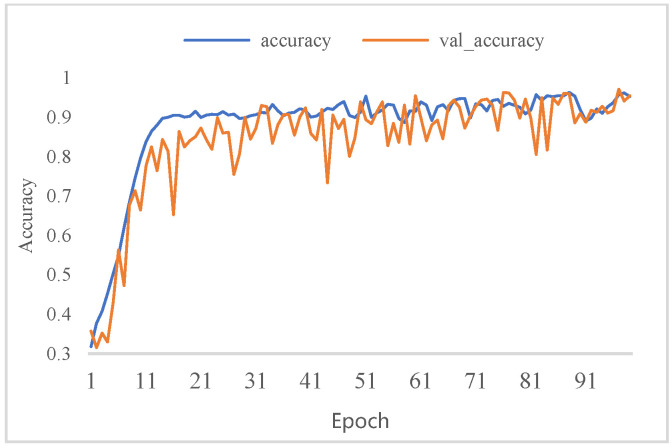
BFPC-Net training set and validation set accuracy.

**Figure 10 micromachines-13-00947-f010:**
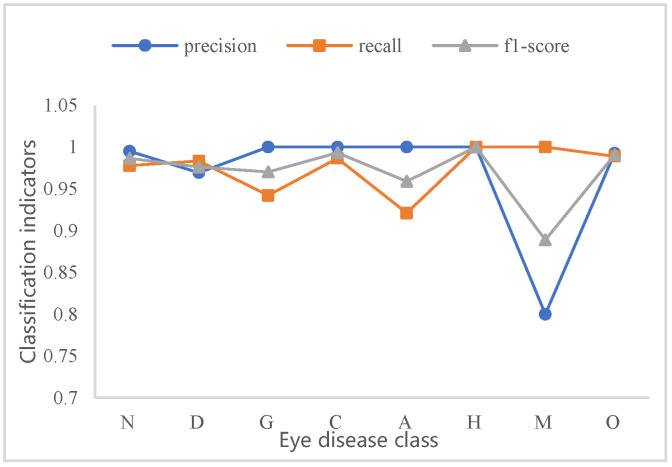
BFPC-Net test classification indicators for each disease.

**Figure 11 micromachines-13-00947-f011:**
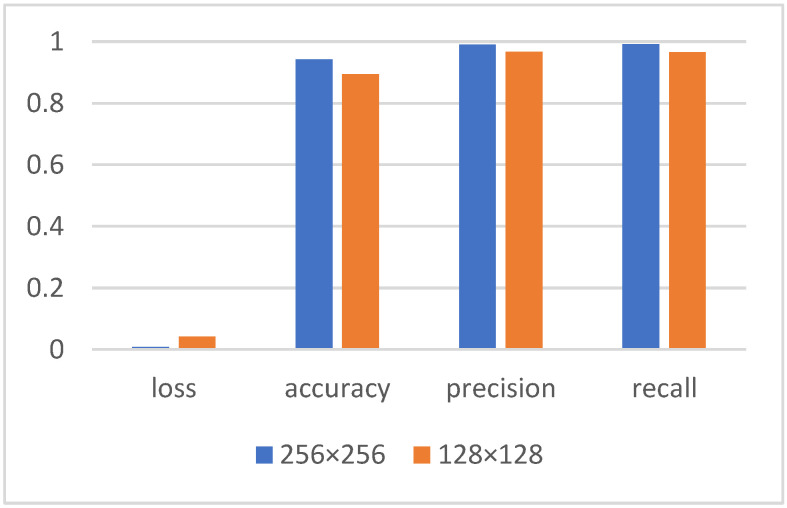
Different image size evaluation metrics for BFPC-Net.

**Table 1 micromachines-13-00947-t001:** Sample distribution after image preprocessing.

	Original Images	Preprocessed Images	Training Set	Test Set
N	2873	2873	2298	575
D	1608	1539	1231	308
G	284	1638	1310	328
C	293	1674	1339	335
A	266	1560	1248	312
H	128	756	605	151
M	232	1206	965	241
O	708	1346	1077	269

**Table 2 micromachines-13-00947-t002:** Comparison of different algorithms (%).

Configuration	Values
Image size	256 × 256
Loss function	Binary crossentropy
Optimizer	Adam
Train/Test	4/1
Epoch	100
Batch size	32
Learning rate	0.001

**Table 3 micromachines-13-00947-t003:** Comparison of different algorithms (%).

Algorithm	Accuracy	Precision	Recall	*F*1 Value
MCGS-Net [28]	-	65.88	61.60	89.66
EfficientNet [20]	92	71	66	89
ResNet [29]	95.47	95.41	94.22	94.75
BFPC-Net	94.23	99.09	99.23	99.16

**Table 4 micromachines-13-00947-t004:** Analysis of ablation experiments for image size 256 × 256 (%).

Models	Accuracy	Precision	Recall	*F*1 Value
Baseline	80.82	83.39	78.36	80.79
Baseline + FFM	94.23	99.09	99.23	99.16

**Table 5 micromachines-13-00947-t005:** Model performance analysis.

Algorithm	Accuracy Rate (%)	*F*1 Value (%)	Number of References (MB)
VGG16 [30]	88.71	88.71	16.29
BFPC-Net	94.23	99.16	12.79

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
