# Peer review of "Multi-Label Fundus Image Classification Using Attention Mechanisms and Feature Fusion"

_micromachines, 2022, doi:10.3390/mi13060947_

Round 1

Reviewer 1 Report

The authors proposed a Multi-label fundus image classification. The work is interesting and needs revision to improve it further. My observations are:

1. The introduction section is missing., All content seems to be merged in the abstract.

2. SOme typos need to fix before the next revision. Structure and refs need to change as per the journal.

3. Literature and introduction both should be different sections. Check your work with following recent studies and cite

"DSCN-net: a deep Siamese capsule neural network model for automatic diagnosis of malaria parasites detection"

"COVID-19 Risk Prediction for Diabetic Patients Using Fuzzy Inference System and Machine Learning Approaches"

"A Non-invasive Approach to Identify Insulin Resistance with Triglycerides and HDL-c Ratio Using Machine learning"

4. Why authors considered BFPC-Net in this work. There are other outperforming models that exist.

5.v Image augmentation model is not clear and understandable.

6. Simulation settings need to write in a table. How your method is outperforming.

7. Result and conclusion sections are poor. More results need to justify to validate the method.

discuss future directions and limitations of the proposed work.

Author Response

Point 1: The introduction section is missing. All content seems to be merged in the abstract.

Response 1: Modify the article structure according to the Journals format, add the Introduction part, and merge it with the Overall part in the original text.

Point 2: SOme typos need to fix before the next revision. Structure and refs need to change as per the journal.

Response 2:

(1) The reference citation format has been updated in accordance with journal format guidelines

Section

Changes

Section 1.1

Gour[14] et al. revise to Gour et al. [14]

glaucoma[12], revise to glaucoma[12]

Section 1.2

Luquan[16] et al. revise to Luquan et al. [16]

J Wang[17] et al. revise to J Wang et al. [17]

Section 1.3

a variety of vision tasks[18] revise to a variety of vision tasks[18]

Section 1.4

Gour et al [14] revise to Gour et al.[14]

(2) The figure reference is revise as follows

Section

Changes

Section 1.3

As indicated in the correlations in Fig. 2 revise to As indicated in the correlations in Figure 2

Section 2

as shown in Fig. 3. revise to as shown in Figure 3

(3)Structural adjustment

The original structure of the article is Overview, Methodology, Results and Discussion, Conclusions, Reference, revise to Introduction, Materials and Methods, Results and Discussion, Conclusions, Reference.

(4)According to MDPI Reference List and Citations Style Guide, the references are revised as follows

In the author's name of the reference 2-17, 19-30, the capitalization of each letter is changed to the capitalization of the first letter of each word.

In references 1-4, 13-20, and 22-30, the reference types after the titles are deleted.

In references 1, 2, 4-11, 13-15, 17-24, 28-30, the ","before and after the journal name is changed to "."

Add the page number after reference 3.

Point 3:. Literature and introduction both should be different sections. Check your work with following recent studies and cite

"DSCN-net: a deep Siamese capsule neural network model for automatic diagnosis of malaria parasites detection"

"COVID-19 Risk Prediction for Diabetic Patients Using Fuzzy Inference System and Machine Learning Approaches"

"A Non-invasive Approach to Identify Insulin Resistance with Triglycerides and HDL-c Ratio Using Machine learning"

Response 3:

(1) Literature is in chapters 1.1-1.4, and the content before the 1.1 chapter is introduction.

(2) The three references mentioned have been added to the 2nd, 3rd and 10th references.

Point 4:. Why authors considered BFPC-Net in this work. There are other outperforming models that exist.

Response 4:

(1)BFPC-Net uses the methods of Image augmentation, Attention mechanisms and Transfer Learning to avoid the problems of model over-fitting and unbalanced input data.

(2)Gour’s research shows that VGG16 pre-trained architecture with SGD optimizer performs better for multi-class multi-label fundus images classification on ODIR database. The AUC and F1 score achieved 84.93 and 85.57, respectively. But the architecture shows an over-fitting phenomenon and class imbalance problems did not solve.

Point 5:. Image augmentation model is not clear and understandable.

Response 5:

In the Image augmentation module, we first highlight the details by image enhancement, and then augment the data set with a small number of images by inversion and rotation. The number of images before and after preprocess is shown in Table 1.

Table 1. Sample distribution after image preprocess.

Origin image

Preprocessed image

training set

test set

N

2873

2873

2298

575

D

1608

1539

1231

308

G

284

1638

1310

328

C

293

1674

1339

335

A

266

1560

1248

312

H

128

756

605

151

M

232

1206

965

241

O

708

1346

1077

269

As shown in added Figure 4 of the paper, the image preprocessing procedure is as follows. As a step in flow chart amplification, rotation and flipping are performed.

Figure 4 Image preprocess flow.

Point 6:. Simulation settings need to write in a table. How your method is outperforming.

Response 6:

(1) Simulation settings

Table 2. Comparison of different algorithms (%).

Configuration

Values

System

Windows 10 pro

GPU acceleration

NVIDIA CUDA 10.2

Framework

Tensorflow

Image size

256×256

Loss function

Binary crossentropy

Optimizer

Adam

Train/Test

4/1

Epoch

100

Batch size

32

Learning rate

0.001

(2) Advantage

In the 8th epoch, the loss has dropped obviously, and the accuracy rate is close to 1. The accuracy difference between the training set and the verification set is small, indicating that the model has not been overfitting.

Point 7:. Result and conclusion sections are poor. More results need to justify to validate the method. Discuss future directions and limitations of the proposed work.

Response 7:

(1) More results

Using images with image sizes of 256×256 and 128×128 for training, the classification results are shown in Figure 11. The results show that, when other conditions are the same, the image is larger and the classification result is better.

Even with 128×128 images as input, the trained accuracy, precision and recall all reach 0.8942, 0.9665 and 0.9660 respectively, which shows that the classification effect of the model for low pixel images is equally excellent.

Figure 11. Different image size evaluate metrics.

(2) future directions and limitations

In the future, more kinds of fundus diseases images can be used to classify, especially rare diseases in clinic. The difficulty of this kind of problem is that the training effect of small sample data classification is poor. However, the classification of such diseases is more suitable for clinical application.

Reviewer 2 Report

The manuscript titled: “Multi-label fundus image classification using attention mechanisms and feature fusion” by Zhen-Wei Li et al., is a valuable contribution to the field of fundus disease classification and deserves publication after minor revisions as follows:

1-    Figure 3 should indicate all the modules described: IAM, RAM and FFM.

2-    Results and discussion should be two separate sections.

3-    The short paragraph at the beginning of the section 3. Results and Discussion should be part of the previous section.

Author Response

Point 1:. Suggest to provide more details of the image augmentation. Why is augmentation needed and how it helps balance the data? Why is the existing augmentation used versus a number of other techniques? What was the original image sizes per class and how many images after augmentation?

Response 1:

(1) As shown in Figure 4 of the paper, the image preprocessing procedure is as follows. As a step in flow image amplification, rotation and flipping are performed.

Figure 4 Image preprocess flow.

(2) As mentioned in Graham and Lopez-Nava, an unbalanced dataset leads to a high misclassification rate and suboptimal performance.

Graham B. Kaggle Diabetic Retinopathy Detection competition report. Technical Report, University of Warwick (2015).

Lopez-Nava, I.H.; Valentín-Coronado, L.M.; Garcia-Constantino, M.; Favela, J. Gait Activity Classification on Unbalanced Data from Inertial Sensors Using Shallow and Deep Learning. Sensors 2020, 20, 4756.

(3) Traditional data augmentation techniques, namely horizontal and vertical flipping and changes in the brightness range, To augment the data set while maximizing the image details, we employ a combination of image enhancement and data augmentation to produce the augmented ODIR dataset.

(4) Table 1 now has add two columns of data, one for the number of images in the original data set and the other for the number of images after preprocessing.

Table 1. Sample distribution after image preprocessing.

Origin image

Preprocessed image

training set

test set

N

2873

2873

2298

575

D

1608

1539

1231

308

G

284

1638

1310

328

C

293

1674

1339

335

A

266

1560

1248

312

H

128

756

605

151

M

232

1206

965

241

O

708

1346

1077

269

Point 2:. For algo comparison (table2), would be good to provide the training time comparison in addition to the accuracy comparison, so that to give readers an idea of model complexities. Also what was the replication numbers those accuracy metrics compared based on? Are there any statistical differences between ResNet and BFPC-Net results? Good to have more details for a full understanding on BFPC-Net.

Response 2:

(1) Due to the difference of equipment, the literature usually does not provide training time as reference, but only provides model parameters as the characteristics of comparative analysis. BFPC-Net’s model parameters is provides in Table 4.

Table 4. Model performance analysis.

Algorithm

Accuracy rate (%)

F1 value (%)

Number of references (MB)

VGG16[30]

88.71

88.71

16.29

BFPC-Net

94.23

99.16

12.79

(2) Replication numbers

In Table 4, the two methods are different in image size and number of training sets. VGG16 uses 224×224 input images, and the number of images used for training is 6,551. BFPC-Net uses 256×256 input images, and the training set is 12,592 images.

(3) Z-test

Z-test is used to test the difference of the average of two groups of samples, so as to judge whether the difference of their respective populations is significant.

Point 3:. Figures 7/8/9 would be good if improved further on the visualization, including providing smaller line sizes, aligning the image fonts with the main text, etc. For Figure7/8 titles, suggest to indicate those are results for the BFPC-Net model.

Response 3:

(1) Figures 7/8/9 line sizes was modified from 2.25 pounds to 0.75 pounds.

(2) Figures 7/8 title change to:

Figure 7. BFPC-Net Ttraining set and validation set losses.

Figure 8. BFPC-Net Ttraining set and validation set accuracy.

Point 4:. About the results on Figure 9, why the M class has a significantly lower f1-score and precision than other classes? Please explain the reason and consider providing more experiments if needed.

Response 4:

Figure 9. BFPC-Net test Classification indicators for each disease.

In ODIR data set, the number of Myopic image set is small, with only 1,206 images after augmentation (show in Table 1). Less image input is not conducive to model training. Hypertension also has a small number of images, but it can be observed that the blood vessel reshape is more obvious in the images, which is beneficial to model learning classification features. However, the features used for classification in Myopic images are not obvious, which leads to the poor classification performance of the model for such diseases.

Point 5:. The formats of equations are suggested to have a major improvement to follow this journal's standard. Positions of equations should align on the same line of main texts (e.g., Equation (1) explanation paragraph); no missing contents (e.g., Equation (3) explanation paragraph).

Response 5:

According to journal's standard, all equations formats are modified, the equations are aligned to the center, and the numbers are aligned to the right. The description of the formula quoted in the article is Equation (x).

Reviewer 3 Report

Overall, the story is sound and paper is well written. I have the below comments for further improving the quality:

1. Suggest to provide more details of the image augmentation. Why is augmentation needed and how it helps balance the data? Why is the existing augmentation used versus a number of other techniques? What was the original image sizes per class and how many images after augmentation? 

2. For algo comparison (table2), would be good to provide the training time comparison in addition to the accuracy comparison, so that to give readers an idea of model complexities. Also what was the replication numbers those accuracy metrics compared based on? Are there any statistical differences between ResNet and BFPC-Net results? Good to have more details for a full understanding on BFPC-Net.

3. Figures 7/8/9 would be good if improved further on the visualization, including providing smaller line sizes, aligning the image fonts with the main text, etc. For Figure7/8 titles, suggest to indicate those are results for the BFPC-Net model.

4. About the results on Figure 9, why the M class has a significantly lower f1-score and precision than other classes? Please explain the reason and consider providing more experiments if needed.

5. The formats of equations are suggested to have a major improvement to follow this journal's standard. Positions of equations should align on the same line of main texts (e.g., Equation (1) explanation paragraph); no missing contents (e.g., Equation (3) explanation paragraph).

Author Response

Point 1: Figure 3 should indicate all the modules described: IAM, RAM and FFM.

Response 1:

Figure 3 title revise

Figure 3. BFPC-Net model. IAM: Image Augment Model, RAM: Residual Attention Module, FFM: Feature Fusion Module.

Point 2: Results and discussion should be two separate sections.

Response 2:

Change the structure of the article, put materials in the second section. The original structure of the article is Overview, Methodology, Results and Discussion, Conclusions, Reference, revise to Introduction, Materials and Methods, Results and Discussion, Conclusions, Reference.

Point 3: The short paragraph at the beginning of the section 3. Results and Discussion should be part of the previous section.

Response 3:

The short paragraph in the third section is revised to "3.1. System Specification", which shows the configuration used in the experiments and the details of the hyperparameters of the model. Add tables to be presented in a more intuitive way.

Table 2. Comparison of different algorithms (%).

Configuration

Values

Image size

256×256

Loss function

Binary crossentropy

Optimizer

Adam

Train/Test

4/1

Epoch

100

Batch size

32

Learning rate

0.001

The results of subsection 3.2.1 shows the basic evaluation metrics of BFPC-Net, which is used to analyze the performance of the model. The results of subsection 3.2.2 shows the performance improvement of the model brought by the module described in section 2. Compared with other methods, the results in subsection 3.2.3 show that this model can achieve higher accuracy in smaller model parameters.

Round 2

Reviewer 1 Report

The article is well revised. I have no further suggestions.